# Porcine Placenta Peptide Inhibits UVB-Induced Skin Wrinkle Formation and Dehydration: Insights into MAPK Signaling Pathways from In Vitro and In Vivo Studies

**DOI:** 10.3390/ijms25010083

**Published:** 2023-12-20

**Authors:** Woo-Jin Sim, Jinhak Kim, Kwang-Soo Baek, Wonchul Lim, Tae-Gyu Lim

**Affiliations:** 1Department of Food Science & Biotechnology, Sejong University, Seoul 05006, Republic of Korea; simjin39@sju.ac.kr; 2R&D Division, Daehan Chemtech Co., Ltd., Gwacheon-si 13840, Republic of Korea; jhkim@dhchemtech.com (J.K.); rnd@dhchemtech.com (K.-S.B.); 3Carbohydrate Bioproduct Research Center, Sejong University, Seoul 05006, Republic of Korea

**Keywords:** porcine placenta, ultraviolet, wrinkle, dehydration, SKH-1, MAPK

## Abstract

Excessive exposure to ultraviolet (UV) radiation from sunlight accelerates skin aging, leading to various clinical manifestations such as wrinkles, dryness, and loss of elasticity. This study investigated the protective effects of porcine placenta peptide (PPP) against UVB-induced skin photoaging. Female hairless SKH-1 mice were orally administered PPP for 12 weeks, followed by UVB irradiation. PPP significantly reduced wrinkle formation, improved skin moisture levels, and prevented collagen degradation. Mechanistically, PPP inhibited the expression of matrix metalloproteinases (MMPs) and upregulated collagen production. Moreover, PPP elevated hyaluronic acid levels, contributing to enhanced skin hydration. Additionally, PPP demonstrated antioxidant properties by increasing the expression of the antioxidant enzyme GPx-1, thereby reducing UVB-induced inflammation. Further molecular analysis revealed that PPP suppressed the activation of p38 MAP kinase and JNK signaling pathways, crucial mediators of UV-induced skin damage. These findings highlight the potential of porcine placental peptides as a natural and effective intervention against UVB-induced skin photoaging. The study provides valuable insights into the mechanisms underlying the protective effects of PPP, emphasizing its potential applications in skincare and anti-aging formulations.

## 1. Introduction

Ultraviolet (UV) radiation from sunlight is categorized into three types: shortwave UV (UVC, 200–290 nm), mediumwave UV (UVB, 290–320 nm), and longwave UV (UVA, 320–400 nm). The Earth’s ozone layer predominantly absorbs UVC radiation, resulting in UVA and UVB being the primary UV types that reach the Earth’s surface. While a substantial amount of UVB radiation reaches human skin mainly around noon, UVA radiation is abundant throughout the day, leading to a significantly higher UVA exposure compared to UVB. However, the biological activity of UVA is considerably weaker in comparison to UVB [1].

Clinical manifestations of photoaging encompass wrinkles, mottled hyperpigmentation, rough skin texture, irregularities in skin tone, dryness, sallowness, deep furrows, severe sagging, telangiectasia, and laxity [2]. Exposure of human skin to UVB radiation induces the release of reactive oxygen species (ROS), activating numerous signaling pathways in keratinocytes and fibroblasts [3,4]. This ultimately leads to the upregulation of various matrix metalloproteinases (MMPs) via transcriptional induction of AP-1, specifically degrading connective tissues like collagen and elastin, destroying extracellular matrix collagen and ceasing new collagen synthesis [5].

UVB-induced skin damage is associated with the generation of ROS and depletion of endogenous antioxidant molecules [6,7,8,9]. In aged skin, a significant depletion of antioxidant enzyme expression occurs in the stratum corneum and epidermis, accumulating oxidatively modified proteins in the upper dermis of photoaged skin [2]. Therefore, oxidative stress induced by UVB plays a crucial role, and intrinsic and extrinsic antioxidants may be necessary to mitigate it.

The placenta serves as a temporary organ in pregnant female mammals, connecting the developing fetus to the uterine wall through the umbilical cord. It is considered a reservoir of cytokines, hormones, physiologically active peptides, enzymes, growth factors, vitamins, and minerals [10]. Extracts from mammalian placentas have been used as ingredients in ointments for skin dryness and beauty [11,12]. Recently, the benefits of localized application of porcine placental peptides for chronic and non-healing wounds have been reported [13,14]. Furthermore, various studies have demonstrated porcine placenta to provide advantages in skin moisturization, elasticity enhancement, immune regulation, and antioxidant effects [15,16,17]. Practical use of porcine placental peptides is considered safe, given that its immune effects are similar to those of the human placenta [16]. However, mechanistic studies on the protective effects of porcine placental peptides against skin photoaging are still warranted.

This study focused on elucidating the mechanism by which porcine placental peptides inhibit skin photoaging induced by UV radiation. Porcine placental peptides effectively reduced skin wrinkles and improved moisture levels in SKH-1 hairless mouse skin photoaging caused by UVB irradiation [18]. We aimed to investigate the potential inhibitory effects of orally administered porcine placenta peptide (PPP) on photoaging in an in vivo model through comprehensive histological analyses, encompassing hematoxylin and eosin staining, Masson’s trichrome staining, and immunohistochemistry. Additionally, we sought to elucidate alterations in the MAPK signaling pathway and various biomarkers, including matrix metalloproteinases (MMPs), hyaluronic synthases (HASs), and cyclooxygenase (COX)-2. This multifaceted approach was undertaken to gain a deeper understanding of the mechanistic basis for the photoaging-inhibitory properties of PPP.

## 2. Results

### 2.1. PPP Inhibits UVB-Induced Skin Wrinkle Formation by Suppressing MMPs Expression

To evaluate the inhibitory effect of PPP against skin wrinkle formation stimulated by UVB, PPP was orally administered to female hairless SKH-1 mice for 12 weeks. During the experiment periods, the mice’s body weight, water intake, and food intake administered with PPP or hyaluronic acid (HA) were not changed (Appendix A). In addition, to answer whether PPP has hepatic toxicity in mice, the levels of hepatic toxicity indicators, including AST and ALT were investigated in the serum of mice. As a result, the PPP-administered group showed no significant changes in serum AST and ALT compared to other groups (Appendix A). Also, the liver, spleen, and kidney weights showed no significant differences among all groups (Appendix A). Before sacrificing the mouse, liquid silica rubber was applied to the dorsal folds, allowed to harden, and the wrinkles modeled on this silica were used as replicas. As shown in Figure 1A, the wrinkle formation in the dorsal skin of the UVB-irradiated group was predominantly presented, while that of the PPP-administered group was markedly decreased. After photographing of the wrinkle formation, the relative roughness average of wrinkle formation was analyzed using Image J 1.54 software. As expected, the relative roughness average of the UVB-irradiated group was significantly higher than that of the normal group. However, the relative roughness average of the PPP- or HA-administered group was lower than that of the UVB-irradiated group (Figure 1A). Several studies reported that collagen is a major component of the skin’s dermal layer and is responsible for preventing skin aging, including wrinkle formation and dehydration [19]. As shown in Figure 1B, the area dyed blue in the skin dermal layer presented an amount of collagen. As quantified results of the collagen intensity by Image J software, the relative collagen intensity of the UVB-irradiated group was lower than that of the normal group, but PPP administration prevented collagen reduction stimulated with UVB in a dose-dependent manner. Simultaneously, to determine whether PPPs affect expression of MMP-1 as collagenase, immunohistochemistry was employed. As shown in Figure 1C, PPP inhibited the amount of MMP-1-positive cells induced by UVB irradiation. As a result of collagen decomposition in the dermal layer by skin exposure to UVB, the epidermal layer becomes thick, and wrinkles are formed while invading the dermis layer [20]. For these reasons, the epidermal thickness was significantly increased in the UVB-irradiated group whereas PPP administration suppressed UVB irradiation and increased epidermal thickness (Figure 1D). As shown in Figure 1E, the protein expression of MMP-1 and MMP-2 was up-regulated by UVB irradiation, whereas PPP administration significantly decreased that of the MMPs. As a substrate of MMPs, several types of collagen mRNA were measured using quantitative real-time PCR. As a result, the transcription level of collagen type VII alpha 1 (*COL7A1*) significantly decreased by UVB irradiation, but PPP protected that of *COL7A1* diminished by UVB in a dose-dependent manner (Figure 1F). These results revealed that PPP inhibited UVB-induced wrinkle formation by suppressing the MMPs expression in SKH-1 hairless mice.

### 2.2. PPP Prevents UVB-Induced Skin Dehydration by Elevating HAS Transcription Level

To measure the effect of PPP on UVB-induced skin dehydration, the transepidermal water loss (TEWL) and relative skin hydration were evaluated at week 12. As shown in Figure 2A,B, UVB irradiation significantly increased the TEWL and decreased the relative skin hydration. However, the TEWL and relative skin hydration alteration were notably recovered by PPP administration. Hyaluronic acid is a hydrophilic molecule with a binding capacity of up to 1000 times its volume in water. Due to this highly potent moisture-absorbing property, it can supply moisture to both the stratum corneum and the dermis, providing skin hydration [21]. In this study, the hyaluronic acid content of the dorsal tissue was decreased by about 0.8-fold by UVB irradiation; however, the contents of the hyaluronic acid irradiated by UVB were recovered by about 0.6-fold by PPP administration (Figure 2C). To investigate the effect of PPP administration on skin moisturizing-related factors, we analyzed the mRNA levels of hyaluronic synthase (*HAS*) *1*, *HAS2*, and long chain base biosynthesis protein 1 (*LCB1*) in dorsal skin tissue using quantitative real-time PCR. As shown in Figure 2D–F, the transcriptional levels of these genes were dose-dependently increased by the PPP administration against UVB irradiation. These results imply that PPP protected the UVB-induced skin dehydration by elevating hyaluronic acid contents in SKH-1 hairless mice.

### 2.3. PPP Reduces UVB-Induced Skin Inflammation by Increasing the Antioxidant Enzyme Transcriptional Level

UV exposure generates reactive oxygen species (ROS) in the epidermis, leading to inflammation, cytokine production, and the formation of wrinkles [22]. As shown in Figure 3A, the administration of PPP increased the mRNA levels of glutathione peroxidase 1 (*GPx-1*), an antioxidant enzyme, in a dose-dependent manner. Moreover, the pro-inflammatory factors induced by UVB irradiation were significantly suppressed by PPP administration (Figure 3B–D). These findings suggest that PPP may function as an antioxidant, playing a role in inhibiting skin wrinkle formation and dryness through its anti-inflammatory effects.

### 2.4. PPP Suppresses UVB-Induced Skin Damage through p38 MAPK Kinase and JNK Signaling Pathways

MAPK signaling cascades are targets for UV and are essential in regulating UV-induced inflammatory reactions [23]. In this study, UVB irradiation significantly activated several MAPK signal molecules, including p38 MAP kinase and JNK, in the dorsal skin of mice. However, PPP notably inhibited the phosphorylation of p38 MAP kinase and JNK stimulated by UVB irradiation (Figure 4). Therefore, these results suggested that PPP may prevent UVB-induced skin damage by targeting p38 MAP kinase and JNK signaling pathways.

### 2.5. PPP Enhances Expression of Collagen in HDF Cells Irradiated with UVB

To validate the suppressive effect of PPP against wrinkle formation in the dorsal skin of mice, the activity and expression of collagenase, which is responsible for inducing skin wrinkle formation by collagen degradation, were measured in HDF cells. PPP did not decrease the viability of HDF cells up to 800 μg/mL (Figure 5A). A treatment concentration of PPP, which exhibited a marginal impact on cell viability within a 10% range, was chosen for the investigation of PPP’s influence on skin hydration. Concentrations of PPP ranging up to 100 μg/mL were utilized in this study to comprehensively explore its effects. As expected, PPP inhibited the collagenase activity and MMP-1 expression in HDF cells irradiated with UVB (Figure 5B,C). Similar to these results, PPP significantly prevented the reduced collagen type III alpha 1 (*COL3A1*) stimulated by UVB irradiation (Figure 5D).

### 2.6. PPP Increases Hyaluronic Acid Contents by Upregulating Transcriptional Factors Associated with Skin Hydration in HDF Cells

To measure the protective effect of PPP against UVB-induced skin dehydration, the hyaluronic acid changes and the mRNA expression of skin hydration-related factors were evaluated in HDF cells. As shown in Figure 6A, PPP significantly enhanced the hyaluronic acid. Additionally, the mRNA expression of the factors associated with skin hydration was significantly increased at PPP treatment concentrations of 50 and 100 μg/mL (Figure 6B–D). These results validated the anti-skin aging effects of PPP in vivo by human-derived cell-based experiments.

### 2.7. Anti-Inflammatory Effect of PPP by Upregulating mRNA Levels of Antioxidant Enzyme

The well-known antioxidants have the potential to prevent UVB-induced cellular inflammation by neutralizing the increased ROS levels [24]. In this study, PPP increased the diminished *GPx-1* mRNA expression stimulated by UVB irradiation in a dose-dependent manner (Figure 7A). In addition, PPP treatment in HDF cells significantly reduced the elevated mRNA expression of pro-inflammatory factors (Figure 7B,C). These results showed that PPP exerts anti-inflammatory action against UVB irradiation by upregulating the expression of antioxidant enzymes.

### 2.8. Effects of PPP on p38 MAPK Kinase and JNK Signaling Pathways in HDF Cells

In this study, PPP suppressed wrinkle formation and dehydration in a UVB-irradiated skin aging mouse model. Similar to these results, PPP significantly decreased the phosphorylation of p38 MAPK kinase and JNK stimulated by UVB irradiation in HDF cells (Figure 8). Therefore, these results imply that PPP has a potential for anti-skin aging against UVB irradiation through upregulating p38 MAPK kinase and JNK signaling pathways in SKH-1 mouse models and human-derived skin cells.

## 3. Discussion

Skin photoaging, a condition characterized by the formation of wrinkles and increased moisture loss in the skin, arises from continuous exposure to ultraviolet radiation in daily life. The porcine placenta has been established as an inner beauty material, supported by scientific evidence from in vivo experiments on UV-induced skin aging and double-masked placebo-controlled clinical studies [16,18]. However, mechanistic studies on the protective effects of porcine placenta against UV-induced skin aging are still warranted.

In this study, SKH-1 hairless mice were orally administered porcine placental peptides and subjected to daily UV radiation for 12 weeks to investigate its protective effects against photoaging. The results revealed a pronounced increase in deep wrinkle formation and moisture reduction in the UV-exposed group compared to the control group without UV exposure after 12 weeks. It is well-known that over 80% of facial skin aging is attributed to UV exposure [25]. Additionally, oral administration of collagen-derived peptides increased the elastin contents in the skin [26]. These studies can support the fact that administered PPP can have a suppression effect on UVB irradiation-induced skin aging. Macroscopic features of skin photoaging include wrinkle formation, rough texture, and loss of skin elasticity. Histological and ultrastructural studies demonstrated epidermal hyperplasia and damage and disruption of collagen fibers [20]. Matrix metalloproteinases (MMPs) constitute a group of endopeptidases reliant on metal ions such as Ca^2+^ and Zn^2+^, secreted by keratinocytes and dermal fibroblasts under various stimuli, including oxidative stress and UV radiation [27]. MMPs play a pivotal role in skin photoaging and can degrade nearly all components of the extracellular matrix (ECM), including collagen, fibronectin, elastin, and proteoglycans.

Additionally, UV radiation is known to induce the expression of several MMPs, with MMP-1 capable of degrading Type I and Type III collagen, the predominant forms of collagen in the dermal layer, and MMP-2 capable of degrading gelatin, a collagen degradation product [28]. In this study, oral consumption of porcine placental peptides significantly inhibited the protein expression levels of UVB-induced MMP-1 and MMP-2 (Figure 1C,E). UVB radiation-induced MMP-1 expression levels were also inhibited by PPP treatment in an in vitro model (Figure 5C). Furthermore, transcriptional levels of *COL7A1*, a gene encoding Type VII collagen, were suppressed by UVB exposure. This indicates that UVB radiation increases the transcription level of *COL7A1*, a common substrate of MMPs, through upregulation of MMP-1 and MMP-2, and such UVB-induced reduction in *COL7A1* was significantly inhibited in mice administered porcine placental peptides (Figure 1F). Meanwhile, Type VII collagen is situated at the physical and biological interface between the epidermis and dermis, playing a pivotal role in providing structural integrity to the dermal-epidermal junction (DEJ) and supporting the structural anchorage between the dermis and epidermis. Destruction of Type VII collagen can lead to the breakdown of the boundary between the epidermis and dermis, potentially contributing to the formation of wrinkles [29]. These findings suggest that porcine placental peptides alleviate primary symptoms of photoaging, such as wrinkle formation, dermal collagen reduction, and epidermal hyperplasia, possibly through inhibiting MMP protein expression (Figure 1A,B,D).

Skin dryness is another prominent symptom of photoaging. Fibroblasts present in the dermis of the skin synthesize and secrete extracellular matrix (ECM) components including collagen and hyaluronic acid (HA) [30,31]. HA, synthesized in the perinuclear region by HA synthase (HAS) 1–3, plays a crucial role in wound healing and tissue repair processes due to its ability to maintain a hydrated environment. Among the HAS enzymes, HAS2 has been identified as the primary isoform in dermal fibroblasts [32]. In this study, oral administration of porcine placental peptides significantly increased the levels of *HAS1* and *HAS2* mRNA following UV radiation exposure, increasing hyaluronic acid content within the skin. Consequently, this inhibited epidermal moisture loss and exhibited moisturizing effects on the skin (Figure 2A–E). An increase in hyaluronic acid levels through the modulation of HAS1 and HAS2 expression highlights a potential mechanism via which porcine placental peptides administration contributes to skin health, providing not only nutraceutical benefits but also supporting essential physiological processes related to skin hydration and repair.

Additionally, sphingolipids play a crucial role in maintaining the integrity and barrier function of the human stratum corneum. LCB1 is one of the key enzymes responsible for synthesizing new sphingolipids, and disruption in sphingolipid formation is considered a significant factor in increasing epidermal moisture loss [33]. In this study, the increase in *LCB1* mRNA resulting from oral administration of porcine placental peptides was consistent with the observed prevention in transepidermal water loss induced by UV irradiation exposure (Figure 2F).

UVB radiation induces excess production of reactive oxygen species (ROS) and depletes endogenous antioxidants such as reduced glutathione, glutathione peroxidase, and catalase in the skin, leading to skin inflammation [34,35]. ROS contributes to activating inflammatory signaling pathways and generating cytokines such as interleukin-1 alpha (IL-1α) and tumor necrosis factor-alpha (TNF-α). Significantly, cytokines like TNF-α induce the production of oxygen radicals and activate or induce the expression of inflammatory enzymes like COX-2, resulting in a mutually reinforcing cycle of oxidative stress and inflammation [36,37]. Thus, ROS and cytokines synergistically potentiate inflammatory skin disorders. In this study, porcine placental peptides increased the concentration-dependent levels of GPx-1 mRNA, which were reduced by UVB exposure (Figure 3A and Figure 7A). Additionally, inflammatory enzyme COX-2 and cytokines levels, which are TNF-α and IL-1α were reduced by PPP in the in vivo (Figure 3B–D) and in vitro (Figure 7B,C) caused by UVB exposure. Considering the synergistic effects of ROS and inflammatory mediators, natural-derived materials with antioxidant and anti-inflammatory properties may represent a promising approach with low toxicity for inhibiting skin damage induced by UVB irradiation [38,39,40].

This study showed that oral administration of porcine placental peptides inhibits the phosphorylation of p38 MAP kinase and JNK induced by UVB in the skin of mice and other organisms (Figure 4). P38 MAP kinase is associated with UVB-induced expression of COX-2 in SKH-1 hairless mouse skin [41]. Furthermore, UVB-induced COX-2 expression in human dermal fibroblasts is regulated by p38 MAP kinase and JNK (Figure 7C and Figure 8). Similarly, it has been reported that pre-treating mouse skin and other tissues with pharmacological inhibitors of p38 MAP kinase and JNK attenuates UVB-induced COX-2 expression [42]. Moreover, it is known that pre-administration of the p38 inhibitor SB242234 orally before UVB exposure blocks the activation of the p38 MAPK cascade, leading to the inhibition of the expression of inflammatory cytokines and enzymes [41].

In conclusion, this study demonstrated that oral intake of porcine placental peptides suppresses skin aging caused by UVB irradiation. These peptides inhibited wrinkle formation and inflammation by attenuating the phosphorylation of p38 MAP kinase and JNK and increasing the levels of factors related to moisturization and antioxidants. These results suggest that porcine placental peptides have excellent wrinkle improvement and moisturizing effects and have potential as a functional food ingredient.

## 4. Materials and Methods

### 4.1. Sample Preparation

The porcine placenta peptide (PPP) powder was provided by Daehan Chemtech Co., Ltd. (Gwacheon-si, Republic of Korea). In brief, the fresh porcine placenta was cleaned, trimmed, and reacted with enzymes for about 4 h. After deactivation of enzymes at high temperatures, the extract was filtered and formed PPP powder through freeze-drying. The provided PPP was dissolved in 0.9% sodium chloride for in vivo studies or dimethyl sulfoxide for in vitro studies and stored at −20 °C until subsequent experiments.

### 4.2. Animal Experiments

Animal experiments were performed according to the guidelines of the Southeast Medi-Chem Institute (approval number: SEMI-23-005, Institutional Animal Care and Use Committee). Five-week-old female SKH-1 hairless mice were used for animal experiments. Mice were housed in a temperature- and humidity-controlled animal facility employing a 12 h light/dark cycle with food and water available ad libitum. After the mice were acclimatized for 1 week, they were randomly divided into 6 groups (*n* = 8): the control group, the UVB-irradiated group, the UVB-irradiated group administrated with PPP 25, 50, or 100 mg/kg b.w., and the UVB-irradiated group administrated with HA 100 mg/kg b.w. as a positive control. The control group and UVB-irradiated group were orally administered 0.9% NaCl, and other groups were orally administered PPP or HA. Oral administration and UVB exposure were three times a week for 12 weeks. UVB (312 nm wavelength) irradiation was gradually increased from 1 minimum erythema dose (MED) (50 mJ/cm^2^) to 4 MED (200 mJ/cm^2^) using UV Crosslinker (UVITEC, Cambridge, UK). Changes in mice’s body weight, water intake, and food intake were evaluated daily for 12 weeks. The TEWL and relative skin hydration were evaluated at week 12 using Corneometer CM825 and Tewameter TM300 equipped with multi probe adapter MPA580 (CK Electronics GmbH, Köln, Germany), respectively. After sacrifice, blood was collected from the inferior vena cava, transferred to a serum-separating tube, incubated at room temperature for 30 min, and centrifuged at 1700× *g* for 15 min. Serum aspartate aminotransferase (AST) and alanine aminotransferase (ALT) were measured using an automatic biochemical analyzer (7100, Hitachi, Tokyo, Japan). Liver, spleen, and kidney tissues were collected and weighed. Dorsal skin tissue was frozen at −80 °C or stored at room temperature in formalin until used for subsequent experiments.

### 4.3. Wrinkle Depth Analysis

Dorsal skin surface replicas of mice were taken using silicon rubber (R201, Biobridge, Hanam, Republic of Korea) before mice were sacrificed at week 12 of the experiment. The replicas were fixed on slide glass and observed using a microscope (Eclipse E600, Nikon, Tokyo, Japan) at 40× magnification. The wrinkle depth of the skin was analyzed using Image J software (National Institutes of Health, Bethesda, MD, USA).

### 4.4. Masson’s Trichrome Staining

Dorsal skin tissue was fixed in 4% buffered formalin. The fixed tissue specimens were embedded in paraffin and sectioned into 4 μm-thick slices. The sections were then de-paraffinized using xylene and sequentially rehydrated with 100% alcohol, 95% alcohol, and 70% alcohol. Masson’s trichrome staining (Abcam, Cambridge, UK) was subsequently applied to visualize tissue structure. The stained dermal collagen area was observed using a microscope and quantified using Image J software.

### 4.5. Immunohistochemistry

Dorsal skin tissue sections were quenched using 0.3% hydrogen peroxide for 5 min, blocked using 3% bovine serum albumin for 30 min, and incubated with a primary antibody against MMP-1 at 4 °C for 16 h. The sections were washed with TBST buffer and incubated with HRP-conjugated secondary antibodies at room temperature for 1 h in the dark. After washing the sections with TBST buffer, the sections were reacted with DAB solution for 5 min. After washing the sections with distilled water, the nucleus was stained with a hematoxylin solution for 5 min. MMP-1-positive cells were observed using a microscope.

### 4.6. Hematoxylin and Eosin Staining

Dorsal skin tissue sections were stained with hematoxylin for 3 min, rinsed with distilled water, differentiated with 0.3% acid alcohol, rinsed with distilled water, and stained with eosin for 2 min. After dehydration, the epidermis and dermis layers were observed using a microscope.

### 4.7. Western Blotting Analysis

Murine dermal tissues, or human dermal fibroblast HDF, were lysed in RIPA buffer at 4 °C for 10 min. The supernatant containing intracellular protein was collected by centrifugation at 12,000× *g* for 10 min, quantified by BCA assay, and heated with sodium dodecyl sulfate (SDS) at 95 °C for 5 min to prepare a loading sample by denaturation into a linear structure. The protein samples were electrophoretically separated by molecular weight through SDS-polyacrylamide gel electrophoresis (SDS-PAGE) at 220 V for 30 min. The separated proteins were electrically transferred to polyvinylidene fluoride (PVDF) membrane at 1.3 A and 25 V for 7 min, and the blotted membrane was blocked with a 5% skim milk solution at room temperature for 1 h. The blocked membrane was incubated with the indicated primary antibody at 4 °C for 16 h. After incubation, the membrane was rinsed with phosphate-buffered saline containing 0.1% tween-20 (PBST) thrice for 10 min, incubated with HRP-conjugated secondary antibody at room temperature for 1 h, and rinsed with PBST. After washing the membrane, the membrane was exposed to ECL solution (Bio-Rad, Hercules, CA, USA) for 1 min, and protein expression was visualized using a chemiluminescence imaging system (ATTO, Tokyo, Japan) at the Biopolymer Research Center for Advanced Materials. The expression level of each protein was analyzed using the Image J software.

### 4.8. Hyaluronic Acid ELISA Assay

Hyaluronic acid contents were measured using Hyaluronan DuoSet ELISA (R&D systems, Minneapolis, MN, USA) according to the manufacturer’s protocol. Briefly, chopped dorsal skin tissue specimens or harvested HDF cells were lysed with RIPA buffer and quantified using the BCA assay. Equal amounts of protein and biotin-labeled antibody were simultaneously added to each well conjugated with the capture antibody against hyaluronic acid and reacted at 37 °C for 45 min. After washing the plate three times with 1× wash solution, HRP-Streptavidin conjugated working solution was added to each well and incubated at 37 °C for 30 min. After washing the plate five times with the wash solution, tetramethylbenzidine substrate was added to each well and reacted at 37 °C in the dark for 15 min. After stopping the reaction by adding stop solution to each well, the absorbance was measured at 450 nm using a Cytation 1 (BioTek, Winooski, VT, USA) at Biopolymer Research Center for Advanced Materials.

### 4.9. Quantitative Real-Time Polymerase Chain Reaction

Total RNA was extracted from dorsal skin tissue and HDF using a TRIzol reagent. The isolated RNA was synthesized into cDNA using the amfiRivert cDNA Synthesis Platinum Master Mix (GenDEPOT, Katy, TX, USA). The primer sequences are shown in Table 1. Quantitative real-time PCR was performed as follows: one cycle of denaturation at 95 °C for 3 min, 40 cycles of denaturation at 95 °C for 15 s, annealing at 58 °C for 15 s, and extension at 72 °C for 30 s. The cycle threshold (Cq) values of each gene were normalized to those of GAPDH. The relative expression of the target gene was compared with that of the control set to 1. Primer sequences are listed in Table 1.

### 4.10. Cell Culture and Sample Treatment

Human dermal fibroblasts were purchased from the American Type Culture Collection (ATCC; Manassas, VA, USA). These cells were cultured in Dulbecco’s modified Eagle’s medium (DMEM; Gibco, Invitrogen, Waltham, CA, USA), including 10% bovine serum albumin (FBS, Invitrogen), 50 U penicillin (Invitrogen), and 50 μg streptomycin (Invitrogen) at 37 °C in a 5% CO_2_ humidified condition.

### 4.11. Treatment of the PPP

The cells were seeded in a 6 cm dish at concentrations of 4 × 10^5^ cells. After incubation for 24 h, the cells were starved with serum-free DMEM. Following incubation for 24 h, the cells were pretreated with PPP at the indicated concentrations for 1 h, irradiated with UVB at 25 mJ/cm^2^, and further incubated for the proper periods suitable for each assay.

### 4.12. Collagenase Activity Assay

A final buffer solution was prepared by dissolving 4 mM calcium chloride in 0.1 M Tris-HCl buffer at pH 7.5. Collagenase from *Clostridium histolyticum* (0.5 mg/mL), substrate 4-phenylazo-benzyloxy-carbonyl-Pro-Leu-Gly-Pro-D-Arg (0.5 mg/mL), and the designated concentration of PPP were diluted in the buffer solution and mixed in a 96-well plate. The plate was subjected to an incubation period of 45 min at 37 °C. To terminate the reaction, 100 μL of 6% citric acid was added. The supernatant’s absorbance was measured at 324 nm using a Cytation 1 (BioTek, Winooski, VT, USA) at Biopolymer Research Center for Advanced Materials.

### 4.13. MTS Assay

To investigate whether PPP affects the viability of HDF cells, the MTS assay was employed. The cells were seeded into a 96-well plate and incubated for 16 h. After incubation, PPP at the indicated concentrations was treated at 37 °C in a humidified 5% CO_2_ condition for 48 h. The cell viability was measured at 490 nm using a Cytation 1 (BioTek, Winooski, VT, USA) at Biopolymer Research Center for Advanced Materials.

### 4.14. Statistical Analysis

All experiments were performed at least three times, and representative results were presented as mean and standard deviation. Statistical significance was indicated using Student’s *t*-test analysis. Results were considered significant when the *p*-value was less than 0.05.

## Figures and Tables

**Figure 1 ijms-25-00083-f001:**
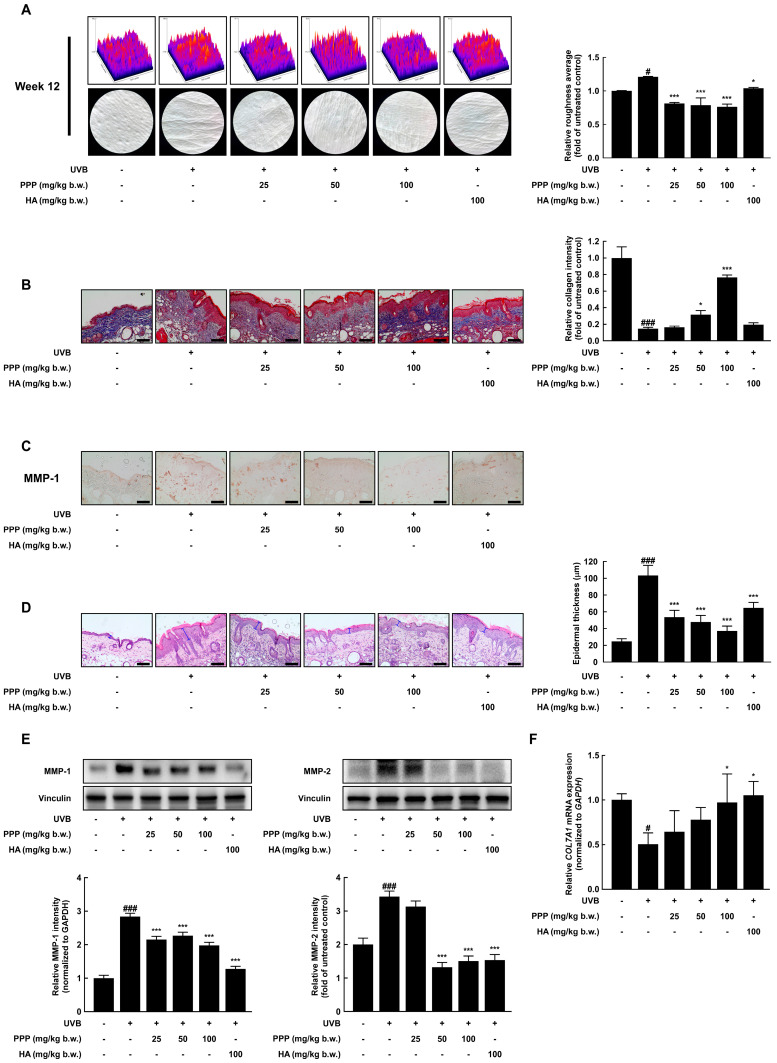
PPP inhibits UVB-induced wrinkle formation in dorsal skin tissue of SKH-1 mice. (**A**) Relative wrinkle formation images obtained from silicon replicas and relative roughness average was calculated by Image J software. (**B**) The collagen of the dermal layer was stained as a blue by Masson’s trichrome staining, and relative collagen intensity was calculated by Image J software. (**C**) The expression of MMP-1 in the dorsal skin was measured by immunohistochemistry. (**D**) Epidermal thickness was observed by H&E staining and calculated by Image J software. The blue bars indicate the epidermis. (**E**) The expression of MMP-1 and MMP-2 in dorsal skin was determined by Western blotting analysis. (**F**) The level of *COL7A1* mRNA was investigated by quantitative real-time PCR. Statistical significance indicated ^#^ *p* < 0.05; ^###^ *p* < 0.001 compared with the untreated group and * *p* < 0.05; *** *p* < 0.001 compared with the UVB-irradiated group.

**Figure 2 ijms-25-00083-f002:**
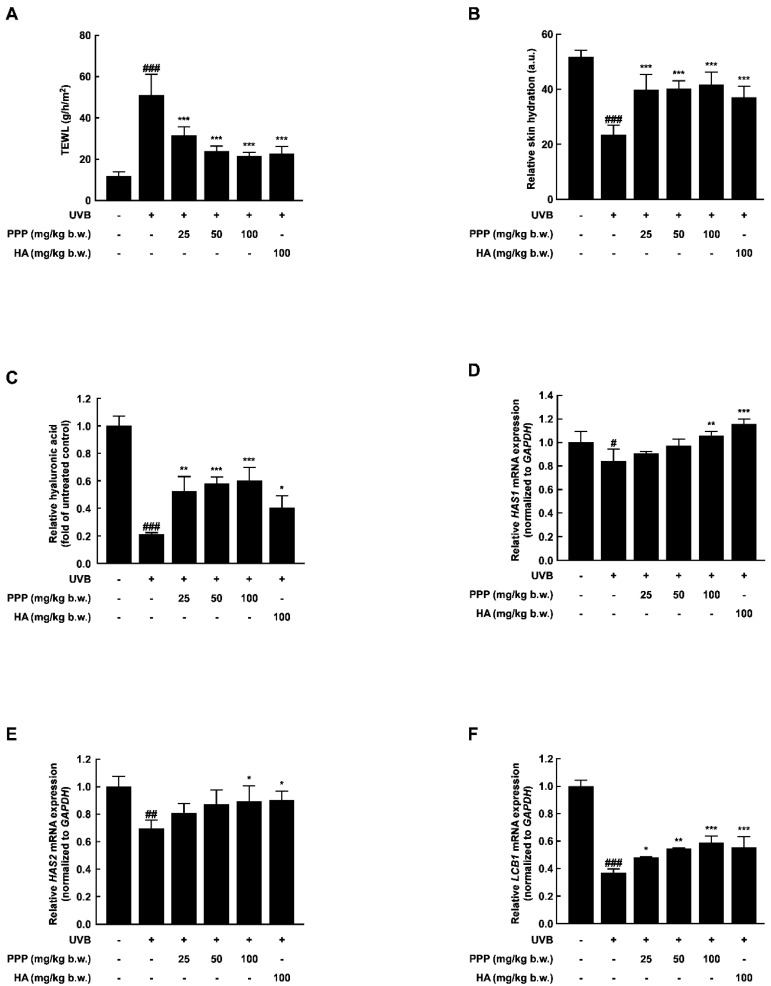
PPP prevents UVB-induced skin dehydration in the dorsal skin tissue of SKH-1 mice. (**A**,**B**) TEWL levels and relative skin hydration of dorsal skin were measured using Corneometer CM825 and Tewameter TM300 equipped with multi probe adapter MPA580. (**C**) ELISA investigated the relative hyaluronic acid of dorsal skin. (**D**–**F**) Quantitative real-time PCR determined the relative mRNA levels of *HAS1*, *HAS2*, and *LCB1*. Statistical significance indicated ^#^ *p* < 0.05; ^##^ *p* < 0.01; ^###^ *p* < 0.001 compared with the untreated group and * *p* < 0.05; ** *p* < 0.01; *** *p* < 0.001 compared with the UVB-irradiated group.

**Figure 3 ijms-25-00083-f003:**
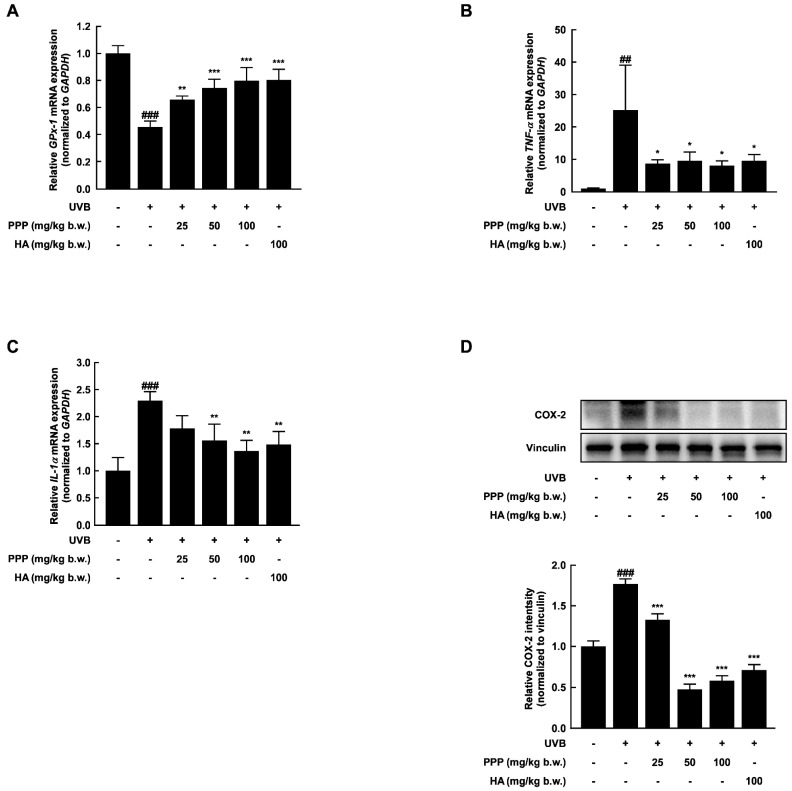
PPP increases the mRNA level of antioxidant enzymes, whereas it reduces the levels of mRNA associated with inflammation. (**A**) The mRNA levels of antioxidant enzyme *GPx-1* and (**B**–**D**) the pro-inflammation factors *TNF-α*, *IL-1α*, and *COX-2* were measured and determined by quantitative real-time PCR. Statistical significance indicated ^##^ *p* < 0.01; ^###^ *p* < 0.001 compared with the untreated group and * *p* < 0.05; ** *p* < 0.01; *** *p* < 0.001 compared with the UVB-irradiated group.

**Figure 4 ijms-25-00083-f004:**
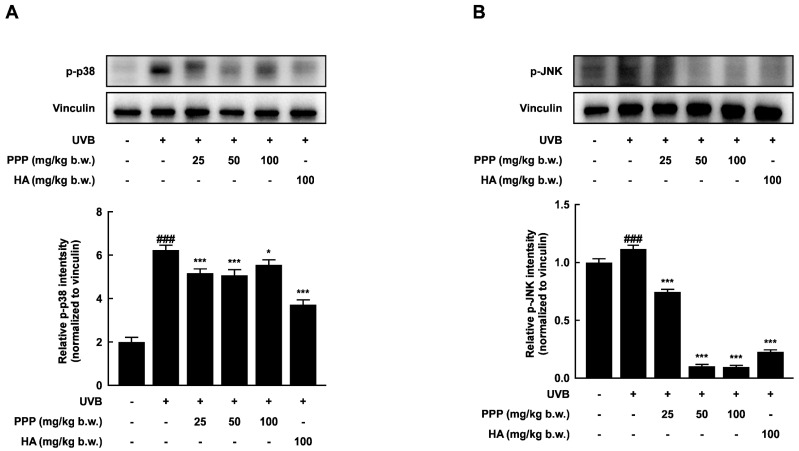
PPP suppresses the activation of p38 MAP kinase and JNK signaling pathways stimulated by UVB irradiation in the dorsal skin tissue of SKH-1 mice. (**A**,**B**) The relative intensity of p-p38 MAP kinase and p-JNK was determined by Western blotting analysis. Statistical significance indicated ^###^ *p* < 0.001 compared with the untreated group and * *p* < 0.05; *** *p* < 0.001 compared with the UVB-irradiated group.

**Figure 5 ijms-25-00083-f005:**
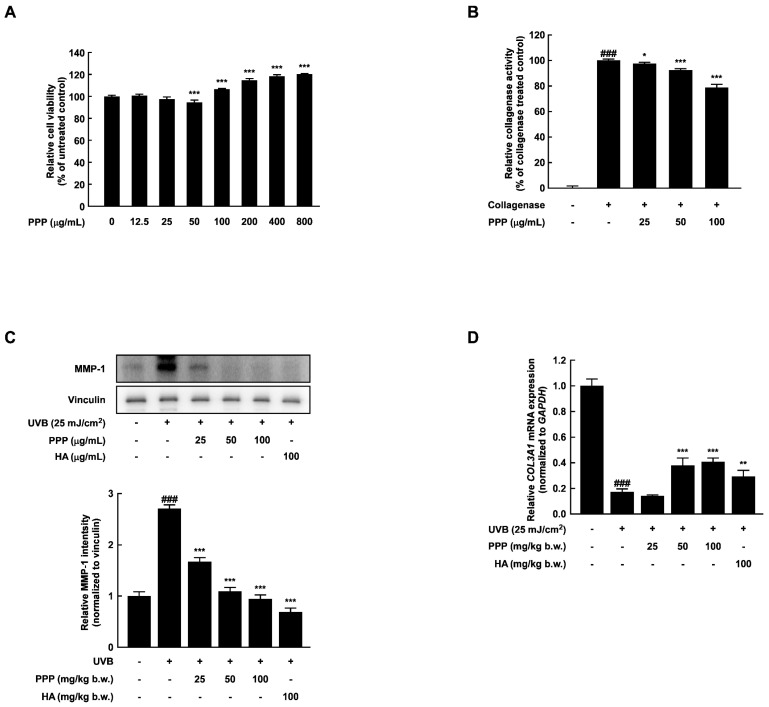
Effect of PPP on activity and expression of collagenase in HDF. The cells were pretreated with PPP at the indicated concentration for 1 h and irradiated with UVB at 25 mJ/cm^2^. After incubation for 24 h, the cells were harvested to evaluate the activity and expression of collagenase. (**A**) The effect of PPP on viability in HDF was assessed by MTS assay. Statistical significance indicated *** *p* < 0.001 compared with the untreated group. (**B**) The effect of PPP on collagenase activity was measured by the collagenase activity manual method. (**C**) The protein level of MMP-1 was investigated by Western blotting analysis. (**D**) Relative *COL3A1* mRNA expression was measured by quantitative real-time PCR. Statistical significance indicated ^###^ *p* < 0.001 compared with the untreated group and * *p* < 0.05; ** *p* < 0.01; *** *p* < 0.001 compared with the UVB-irradiated group.

**Figure 6 ijms-25-00083-f006:**
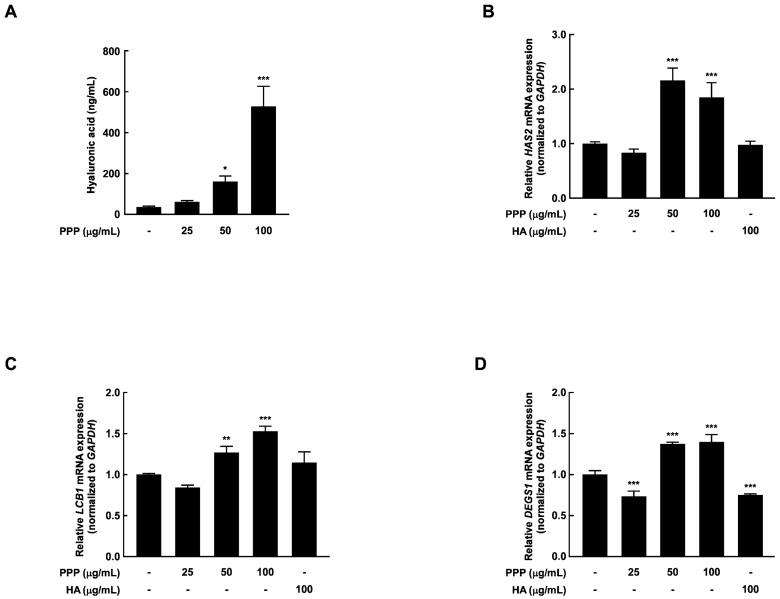
Effect of PPP on hydration in HDF. The cells were pretreated with PPP at the indicated concentration for 1 h, irradiated with UVB at 25 mJ/cm^2^, and further incubated for 24 h. (**A**) Hyaluronic acid was measured by ELISA assay. After incubation for 24 h, the cell supernatant was collected for measuring the extracellular hyaluronic acid content. (**B**–**D**) After incubation for 24 h, the cell lysate was harvested and used for total RNA extraction. The mRNA expression related to skin hydration was determined by quantitative real-time PCR. Statistical significance indicated * *p* < 0.05; ** *p* < 0.01; *** *p* < 0.001 compared with the untreated group.

**Figure 7 ijms-25-00083-f007:**
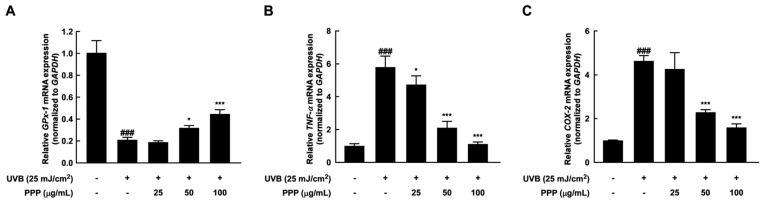
Effect of PPP on expression of the mRNA associated with antioxidant and anti-inflammation in HDF. The cells were pretreated with PPP at the indicated concentration for 1 h, irradiated with UVB at 25 mJ/cm^2^, and further incubated for 24 h. After incubation, the cell lysate was collected and used for total RNA extraction. (**A**) The expression of *GPx-1* mRNA and (**B**,**C**) the expression of mRNA involved in pro-inflammation were determined by quantitative real-time PCR. Statistical significance indicated ^###^ *p* < 0.001 compared with the untreated group and * *p* < 0.05; *** *p* < 0.001 compared with the UVB-irradiated group.

**Figure 8 ijms-25-00083-f008:**
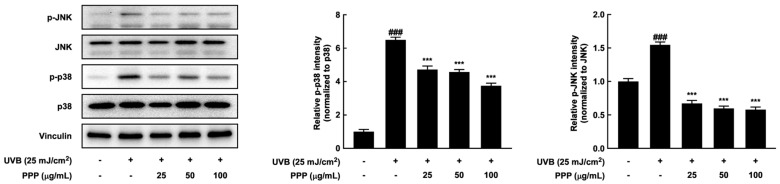
Effect of PPP on phosphorylation of p38 MAP kinase and JNK in HDF. The cells were pretreated with PPP at the indicated concentration for 1 h, irradiated with UVB at 25 mJ/cm^2^, and further incubated for 3 h. After incubation, the cells were lysed to determine the indicated protein expression. Statistical significance indicated ^###^ *p* < 0.001 compared with the untreated group and *** *p* < 0.001 compared with the UVB-irradiated group.

**Table 1 ijms-25-00083-t001:** Sequences of the primers.

Gene	Organism	Primer Sequences (5’ → 3’)	Accession Number
GAPDH	M	F: CAT CAC TGC CAC CCA GAA GAC TG R: ATG CCA GTG AGC TTC CCG TTC AG	NM_001411843.1
COL7A1	M	F: CAG GCA TTG GTG CCA GTG AAC A R: TGG ACC TCC TAC CTC ACA GTC A	NM_007738.4
HAS1	M	F: GTG CGA GTG TTG GAT GAA GAC C R: CCA CAT TGA AGG CTA CCC AGT ATC	NM_008215.2
HAS2	M	F: GCC ATT TTC CGA ATC CAA ACA GAC R: CCT GCC ACA CTT ATT GAT GAG AAC C	NM_008216.3
LCB1	M	F: AGC GCC TGG CAA AGT TTA TG R: GTG GAG AAG CCG TAC GTG TAA AT	NM_009269.2
Gpx-1	M	F: CCC ACT GCG CTC ATG A R: GGC ACA CCG GAG ACC AAA	NM_001329528.1
TNF- α	M	F: GGT GCC TAT GTC TCA GCC TCT T R: GCC ATA GAA CTG ATG AGA GGG AG	NM_001278601.1
IL-1α	M	F: TCG CAG CAG GGT TTT CTA GG R: CAG CTT TAA GGA CGG GAG GG	NM_010554.4
GAPDH	H	F: GTC TCC TCT GAC TTC AAC AGC G R: ACC ACC CTG TTG CTG TAG CCA A	NM_001357943.2
COL3A1	H	F: TGG TGC CCC TGG TCC TTG CT R: TAC GGG GCA AAA CCG CCA G	NM_000090.4
HAS2	H	F: TCG CAA CAC GTA ACG CAA T R: ACT TCT CTT TTT CCA CCC CAT TT	NM_005328.3
LCB1	H	F: CCA TGG AGT GGC CTG AAA GA R: CTG ACA CCA TTT GGT AAC AAT CCT A	NM_001281303.2
DEGS1	H	F: GCT GAT GGC GTC GAT GTA GA R: TGA AAG CGG TAC AGA AGA ACC A	NM_001321542.2
Gpx-1	H	F: TTC CCG TGC AAC CAG TTT G R: GGA CGT ACT TGA GGG AAT TCA GA	NM_001329503.2
TNF -α	H	F: ATC CTG GGG GAC CCA ATG TA R: AAA AGA AGG CAC AGA GGC CA	NM_000594.4
COX-2	H	F: CGG TGA AAC TCT GGC TAG ACA G R: GCA AAC CGT AGA TGC TCA GGG	NM_000963.4

## Data Availability

All the data are contained within the article and Appendix A.

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
