# Peer review of "Porcine Placenta Peptide Inhibits UVB-Induced Skin Wrinkle Formation and Dehydration: Insights into MAPK Signaling Pathways from In Vitro and In Vivo Studies"

_ijms, 2023, doi:10.3390/ijms25010083_

Round 1

Reviewer 1 Report

Comments and Suggestions for Authors

The manuscript „Porcine Placental Peptide Inhibits UVB-Induced Skin Wrinkle Formation and Dehydration: Insights into MAPK Signaling Pathways from In Vitro and In Vivo Studies” presents a study on porcine placental product use in potential anti-aging treatment.

The rejuvenation, protection and reversal of photodamage to the skin are intensively studied, making this topic important and interesting. Unfortunately, the presentation of the preformed research is, in my opinion, not sufficient for publication.

1.      The product used in not a PEPTIDE, it is probably a mixture of peptides formed by enzymatic digestion of placenta. The description in the Materials and Methods is not sufficient. Either the commercial product should be specified, or the whole procedure described in detail. At the moment the description does not allow the repetition of the experiment and the name used in the paper is misleading.

2.      PPP was dissolved in DMSO and stored at low temperature. How were the samples prepared for oral application and cell tests? Was there a control for DMSO performed, as it may affect immune response and other investigated effects?

3.      There is quite a lot of research on placental peptides and placental preparations, the information in the introduction is limited, but even the mentioned examples are not scarce (as far as safety is concerned) (line 59-60).

4.      In lines 65-70 (Introduction), results are introduced, not the aim of the study. The Results or the end of Introduction should contain a clear plan of the research (animal study and cell line study).

5.      In Results, there are several abbreviations without immediate explanation, or the explanation appears in Methods (HA, .

6.      The description of experiments in results is not sufficient or not clear enough (line 82: At the end of the animal experiment, the dorsal wrinkle was imitated with a silica rubber replica, line 86: the relative roughness average of wrinkle formation was analyzed using Image J software (probably after scanning or photographing?))

7.      Collagen is not the only factor responsible for wrinkle formation (neural tone in facial muscles?), although it is responsible for skin condition and age-related changes. Collagen-derived peptides and collagen may act differently.

8.      Figure 1: panel B – the first picture differs from others, despite scale bar, please comment.  panel D – how was the measurement place selected (last picture?)

9.      In general, the Authors regularly use a phrase (in a dose-dependent manner). The bar graphs show in some cases lack of relation (panel E in figure 1, 6B,C) or the dependence is weak (small changes, like in 2C). The flat rate suggests either strong effect of a small dose, or limited effect.

10.   The effects analyzed for animal tests and cell culture should be presented in comparable way (TNF etc.) and discussed.

11.   There is no discussion on oral administration of peptide products (digestion etc.) and their transport to the respective tissues.

12.   Please clarify the information: (line 185) PPP did not decrease the viability of HDF cells up to 800 μg/mL and (line 188) as the concentration affecting cell viability under 10%.

13.   Discussion contains a lot of material that is more suitable in Introduction, whereas the obtained results are just mentioned, not confronted or actually discussed.

14.   Methods: in what form was PPP administered, did the control groups obtain placebo food? Was the administration daily? How long was daily irradiation?

15.   “Dorsal skin tissue was collected and stored at –80°C until used for subsequent experiments.” Did the Authors investigate the effect of deep freezing on skin properties? Are there references for such treatment? At which step were the silicone prints made?

16.   For materials, references to detailed procedures and origin and preparation of reagents are needed. (Phrase in line 400: Hyaluronic acid contents were measured according to the manufacturer’s protocol – without reference it is not acceptable).

17.   Please specify the control experiments for HDF.

18.   In 4.11 there is no mention of PPP (only RPE??), please clarify the reason for this test.

19.   In general, the effects of PPP after oral administration and HDF should be discussed and the statements in Conclusions re-considered. The presented effects are in several cases limiting the damage, not preventing it (which may suggest complete reversal).

Minor issues:

In Abstract, the abbreviations are different than in the text (U.V. v. UV etc.)

Line 98: style: Simultaneously, to whether PPP affect expression of MMP-1 as collagenase, immunohistochemistry was employed.

Line 104: PPP administration was suppressed the epidermal thickness (?)

Line 264: Ca2+

Comments on the Quality of English Language

The language in some cases is artificailly complex. 

Author Response

Reviewer #1

Comment 1. The product used in not a PEPTIDE, it is probably a mixture of peptides formed by enzymatic digestion of placenta. The description in the Materials and Methods is not sufficient. Either the commercial product should be specified, or the whole procedure described in detail. At the moment the description does not allow the repetition of the experiment and the name used in the paper is misleading.

Response: We appreciate the reviewer's kind indication. The PPP is not a single PEPTIDE. PPP is a peptides mixture, and we indicated representative peptides sequences in the Materials and Methods section (lines 360––362).

Comment 2. PPP was dissolved in DMSO and stored at low temperature. How were the samples prepared for oral application and cell tests? Was there a control for DMSO performed, as it may affect immune response and other investigated effects?

Response: We appreciate the reviewer's kind comments. DMSO may affect the immune response, as the reviewer's comment suggests. In animal studies, the PPP was dissolved in 0.9% sodium chloride and orally administered. Before oral administration, PPP was stored at room temperature for 30 minutes. In vitro studies, PPP was dissolved in DMSO and prepared at room temperature for approximately 30 min. Additionally, to avoid DMSO affecting the cells, cells were allowed to be treated with DMSO at a concentration of 0.1% or less. So, the explanation of PPP preparation was additionally described in the sample preparation section of the revised manuscript (line 362).

Comment 3. There is quite a lot of research on placental peptides and placental prepartions, the information in the Introduction is limited, but even the mentioned examples are not scarce (as far as safety is concerned) (line 59–60)

Response: We appreciate the reviewer's kind indication. We agree with your comments and have revised the sentence (lines 59–60).

Comment 4. In Lines 65–70 (Introduction), results are introduced, not the aim of the study. The Results or the end of the Introduction should contain a clear plan of the research (animal study and cell line study).

Response: We appreciate your kind comment. Based on reviewer comments, we revised the Introduction to include a clear plan for the study (line 66–74).

Comment 5. In Results, there are several abbreviations without immediate explanation, or the explanation appears in Methods (HA, .)

Response: We appreciate your kind indication. Based on reviewer comments, we have revised or added explanations of abbreviations in the whole revised manuscript as follows:

In lines 11, 13, 17, 80, 111, 139, 140, 157, and 195: ultraviolet (UV), porcine placental peptide (PPP), matrix metalloproteinases (MMPs), hyaluronic acid (HA), collagen type VII alpha 1 (COL7A1), hyaluronic synthase (HAS) 1, long chain base biosynthesis protein 1 (LCB1), glutathione peroxidase 1 (GPx-1), collagen type III alpha 1 (COL3A1)

Comment 6. The description of experiments in results is not sufficient or not clear enough (line 82) At the end of the animal experiment, the dorsal wrinkle was imitated with a silica rubber replica, line 86: the relative roughness average of wrinkle formation was analyzed using image J software (probably after scanning or photographing?)

Response: We appreciate your kind indication. As per the indication, we revised the description of experiments in the Results section.

In lines 85–87: "Before sacrificing the mouse, liquid silica rubber was applied to the dorsal folds, allowed to harden, and the wrinkles modeled on this silica were used as replicas. Replicas were imaged and analyzed with Image J software."

In line 89: change from "observation" to "photographing"

Comment 7. Collagen is not the only factor responsible for wrinke formation (neural tone in facial muscles?), although it is responsible for skin condition and age-related changes. Collagen-derived peptides and collagen may act differently.

Response: This study focuses on the protective effects of PPP against the representative wrinkle formation and moisture reduction in photoaging caused by ultraviolet (UV) radiation. The skin acquires its structural and unique characteristics through a network of fibrous proteins such as collagen and elastin with extracellular matrix (ECM). Wrinkle formation is a well-known phenomenon resulting from the degradation of collagen, a structural and functional element in the skin's dermis, along with ECM, due to enzymatic actions such as increased matrix metalloproteinases (MMPs) induced by UV exposure. Moreover, MMP-1 plays a role in breaking down collagen as a substrate. Based on this, it is speculated that PPP inhibits collagen degradation caused by UV-induced skin photoaging by suppressing MMP-1. This inhibition is presumed to hinder wrinkle formation significantly.

Comment 8. Figure 1: panel B–the first picture differs from others, despite scale bar, please comment. Panael D – how was the measurement place selected (last picture?)

Response: We appreciate the reviewer's kind indication. In Figure A, the first picture size is the same as the other. UVB irradiation increased the epidermal thickness, which may be why they look different in size in pictures. In all photographs, the places where dermal collagen fibers could be well seen were selected among the skin tissues of each group.

Comment 9. In general, the Authors regularly use a phrase (in a dose-dependent manner). The bar graphs show in some cases lack of relation (panel E in figure 1, 6B, C)or the dependence is weak (small changes, like in 2C). The flat rate suggests either strong effect of a small dose, or limited effect.

Response: We appreciate the reviewer's kind comment. As per the reviewer's comment, we revised the previous sentence to be more accurately expressed as follows:

In lines 106–109: "PPP administration suppressed UVB irradiation-increased epidermal thickness (Figure 1D). As shown in Figure 1E, the protein expression of MMP-1 and MMP-2 was up-regulated by UVB irradiation, whereas PPP administration significantly decreased that of the MMPs."

In lines 135–138: "In this study, the hyaluronic acid content of the dorsal tissue was decreased by about 0.8-fold by UVB irradiation; however, the contents of the hyaluronic acid irradiated by UVB were recovered by about 0.6-fold by PPP administration (Figure 2C)."

In lines 213–214: Moreover, the mRNA expression of the factors associated with skin hydration was sig-nificantly increased (Figure 6B–D).

Comment 10. The effects analyzed for animal tests and cell culture should be presented in comparable way (TNF etc.) and discussed.

Response: We appreciate the reviewer's kind comment. We discussed and compared the effects analyzed in animal models and cell culture models in the discussion section of the revised manuscript as follows:

In lines 285–286: "UVB radiation-induced MMP-1 expression levels were also inhibited by PPP treatment in an in vitro models (Figure 5C)."

In lines 329–332: "In this study, porcine placental peptide increased the concentration-dependent levels of GPx-1 mRNA, which were reduced by UVB exposure (Figure 3A and 7A). Additionally, inflammatory enzyme COX-2 and cytokines levels, which are TNF-α and IL-1α were reduced by PPP in the in vivo (Figure 3B–D) and in vitro (Figure 7B and C)."

Comment 11. There is no discussion on oral administration of peptide products (digestion etc.) and their transport to the respective tissues.

Response: We appreciate the reviewer's kind suggestion. As per the reviewer's comment, we discussed the digestion and anti-skin aging effects of peptides in the discussion section of the revised manuscript (lines 268–272) as follows: "Previous studies have reported that orally administered collagen-derived peptides are digested and detected in the circulatory system [26]. Additionally, oral administration of collagen-derived peptides increased the elastin contents in the skin [27]. These studies can support the fact that administered PPP can have a suppression effect on UVB irradiation-induced skin aging."

Comment 12. Please clarify the information: (line 185) PPP did not decrease the viability of HDF cells up to 800 μg/mL and (line 188) as the concentration affecting cell viability under 10%.

Response: We appreciate the reviewer's kind comment. As per the reviewer's comment, we have revised the sentence (lines 189–192) as follows: "A treatment concentration of PPP, which exhibited a marginal impact on cell viability within a 10% range, was chosen for the investigation of PPP's influence on skin hydration. Concentrations of PPP ranging up to 100 μg/mL were utilized in this study to comprehensively explore its effects."

Comment 13. Discussion contains a lot of material that is more suitable in Introduction, whereas the obtained results are just mentioned, not confronted or actually discussed.

Response: We appreciate the reviewer's comment. As per the reviewer's comment, we revised the discussion section of the revised manuscript as follows:

In lines 309–313: "An increase in hyaluronic acid levels through the modulation of HAS1 and HAS2 expression highlights a potential mechanism via which porcine placental peptide administration contributes to skin health, providing not only nutraceutical benefits but also supporting essential physiological processes related to skin hydration and repair."

Comment 14. Methods: in what form was PPP administered, did the control groups obtain placebo food? Was the administration daily? How long was daily irradiation?

Response: We appreciate the reviewer's comment. As per the reviewer's comment, we have added the administered information between groups, the administration and UVB exposed cycle, and the house condition (section 4.2.) as follows:

In lines 369–370 and 374–376: "Mice were housed in a temperature- and humidity-controlled animal facility employing a 12 h light/dark cycle with food and water available ad libitum." and "The control group and UVB-irradiated group were orally administered 0.9% NaCl, and other groups were orally administered PPP or HA. Oral administration and UVB exposure were three times a week for 12 weeks."

Comment 15. "Dorsal skin tissue was collected and stored at –80°C until used for subsequent experiments." Did the Authors investigate the effect of deep freezing on skin properties? Are there references for such treatment? At which step were the silicone print made?

Response: We appreciate the reviewer's kind indication. The collected skin tissues were stored in two forms: at room temperature in 10% formalin for histological analysis and frozen at –80°C for RNA and protein analysis. Tissues intended for analysis of RNA and protein expression but not for histological analysis are generally stored at –80°C. The replicas of dorsal skin tissue were prepared before mice were sacrificed at week 12 of the experiment. We have added the tissue storage and replica preparation conditions in the revised manuscript (sections 4.2. and 4.3.) as follows:

In lines 386–388: "Dorsal skin tissue was frozen at –80°C or stored at room temperature in formalin until used for subsequent experiments."

In lines 390–391: "Dorsal skin surface replicas of mice were taken using silicon rubber (R201, Biobridge, Hanam, Republic of Korea) before mice were sacrificed at week 12 of the experiment."

Comment 16. For materials, references to detailed procedures and origin and preparation of reagents are needed. (Phrase in line 400: Hyluronic acid contents were measured according to the manufacturer's protocol –without reference it is not acceptable).

Response: We appreciate the reviewer's comment. As per the reviewer's comment, we have indicated the manufacturing company information in the revised manuscript Materials and Methods section 4.8. In lines 435–436.

Comment 17. Please specify the control experiments for HDF.

Response: We appreciate the reviewer's indication. Based on reviewer comments, we have separated the control experiments for HDF in the Materials and Methods section 4.11.

Comment 18. In 4.11 there is no mention of PPP (only PPE?), please clarify the reason for this test.

Response: We appreciate the reviewer's indication. We confirmed an error in the sample labeling and revised it from PPE to PPP in line 471.

Comment 19. In general, the effects of PPP after oral administration and HDF should be discussed and the statements in Conclusions re-considered. The presented effects are in several cases limiting the damage, not preventing it (which may suggest complete reversal).

Response: We appreciate the reviewer's comment. We considered the reviewer's comment and added the modified conclusion in the discussion section of the revised manuscript as follows:

In lines 347–353: "In conclusion, this study demonstrated that oral intake of porcine placental peptide suppresses skin aging caused by UVB irradiation. This resulted in regulating wrinkle formation, moisturizing, inflammation, and antioxidant factors by attenuating the phosphorylation of p38 MAP kinase and JNK. Therefore, there is a need to secure clinical evidence in the future, and based on the preclinical research results, we suggest that porcine placenta peptides, which have excellent wrinkle improvement and moisturizing effects, have potential as functional food ingredients."

Minor Issues

In the abstract, the abbreviations are different than in the text (U.V. v. UV, etc.).

Line 98: style: Simultaneously, to determine whether PPP affects the expression of MMP-1 as a collagenase, immunohistochemistry was employed.

Line 104: PPP administration suppressed the epidermal thickness (?)

Line 264: Ca2+

Response: We appreciate the reviewer's kind indication. We confirmed the sentence and revised the contents.

Reviewer 2 Report

Comments and Suggestions for Authors

Further insight into the PPP sample preparation, specifically about the accessibility, resource outlook, and shelf life would add value to the introduction section. The results section of the manuscript is lengthy and could be shortened by moving the reasoning behind the chosen tests and assays into the methods section, and leading with the methods section, before transitioning into the results section. Within the methods, the UVB radiation technique should be explained in full, including the actual device used, while elaborating on the specs. Would also be insightful to include the numbers of mice in each cohort. The conclusion, while adequate, could be lengthened by further elaborating on the impact this study has on the topic of PPP and a suggestion for the direction of further research on this topic.

Author Response

Reviewer #2

Comment 1. Further insight into the PPP sample preparation, specifically about the accessibility, resource outlook, and shelf life would add value to the introduction section.

Response: We appreciate the reviewer's kind suggestion. We already introduced the porcin planceta contents in the introduction section (lines 51–62). However, we agreed to lack the information from the sample preparation. We added detailed information to the materials and methods section of the revised manuscript.

Comment 2. The results section of the manuscript is lengthy and could be shortened by moving the reasoning behind the chosen tests and assays into the methods section, and leading with the methods section, before transitioning into the results section.

Response: We appreciate the reviewer's kind suggestion. As per the reviewer's suggestion, the method contents were transferred from the results section to the materials and methods section. The sentences " using Corneometer CM825 and Tewameter TM300 equipped with Multi probe adapter MPA580 (CK Electronics GmbH, Köln, Germany), respectively." was inserted into lines 378–381.

Comment 3. Within the methods, the UVB radiation technique should be explained in full, including the actual device used, while elaborating on the specs.

Response: We appreciate the reviewer's kind indication. As per the reviewer's comment, we added the UVB irradiation condition and information in lines 376 and 378 of the materials and methods section 4.2.

Comment 4.Would also be insightful to include the numbers of mice in each cohort.

Response: We appreciate the reviewer's kind indication. As per the reviewer's comment, we added the information about the mouse number in line 371 of the materials and methods section 4.2.

Comment 5. The conclusion, while adequate, could be lengthened by further elaborating on the impact this study has on the topic of PPP and a suggestion for the direction of further research on this topic.

Response: We appreciate the reviewer's kind suggestion. As per the reivewer's comment, we revised the conclusion as follows:

In lines 347–353: "In conclusion, this study demonstrated that oral intake of porcine placental peptide suppresses skin aging caused by UVB irradiation. This resulted in regulating wrinkle formation, moisturizing, inflammation, and antioxidant factors by attenuating the phosphorylation of p38 MAP kinase and JNK. Therefore, there is a need to secure clinical evidence in the future, and based on the preclinical research results, we suggest that porcine placenta peptides, which have excellent wrinkle improvement and moisturizing effects, have potential as functional food ingredients."

Round 2

Reviewer 1 Report

Comments and Suggestions for Authors

1. I would like to bring to attention of the Authors the part of first review, which, in my opinion, was not sufficiently addressed:

"The product used is probably a mixture of peptides formed by enzymatic digestion of placenta. The description in the Materials and Methods is not sufficient. Either the commercial product should be specified, or the whole procedure described in detail. At the moment the description does not allow the repetition of the experiment and the name used in the paper is misleading."

1A. In the current version the investigated product is still called a peptide, although the Authors themselves state that it is a mixture of peptides.

Such a misleading name cannot be used.

1B. The source or method of obtaining the analysed preparation - is this a commercial product, a product being developed? Was the powder obtained from the Daehan company already after the enzyme treatment? Adding peptide sequences opens a new set of questions about quantification, methods of analysis and standarization of PPP.
The current description leaves a doubt what was done by the company and what was done by the Authors. Leaving unspecified procedure in product description makes it not clear and makes replicability doubtful.

1C. There may be a difference between the PPP and its composition after digestion (compare line 268, ref 26). This may suggest that although the preparation investigated in vivo and in vitro is the same, the substances causing the observed effects are not the same. I would suggest more caution with cause-effect statements in this paper.

1D: example:

"In this study, porcine placenta peptide increased the concentration- dependent levels of GPx-1 mRNA, which were reduced by UVB exposure"
Such sentence suggests this is a single peptide (if read separately), which is not studied in this work (a mixture is used).

2. Please either provide reference for the sentence in lines 64-66, or modify it to relate it to the results of this work.

3. Please check the dose dependence in line 193.

4. In figure 6B-D, the effect of 25 microgram/mL dose and higher doses needs to be addressed, as it is not straightforward.

5. Please consider modification in line 349 - inflammation (negative effect) is next to positive effects, which may suggest that the preparation in increasing inflammation (the direction of regulation).

Line 351: are the preclinical studies refering to this work, or other results? This should be specified and, if the preclinical character is assigned, mentioned in the text earlier.

Author Response

Reviewer #1

Comment 1A. In the current version the investigated product is still called a peptide, although the Authors themselves state that it is a mixture of peptides.

Response: We appreciate the reviewer's kind indication. As per the reviewer's comment, all content in the manuscript was changed from “porcine placental peptide” to “porcine placental peptides.”

Comment 1B. The source or method of obtaining the analyzed preparation – is this a commercial product, a product being developed? Was the powder obtained from the Daehan company already after the enzyme treatment? Adding peptide sequences opens a new set of questions about quantification, methods of analysis and standardization of PPP.

The current description leaves a doubt what was done by the company and what was done by the Authors. Leaving unspecified procedure in product description makes it not clear and makes replicability doubtful.

Response: We appreciate the reviewer's kind suggestion. Daehan Chemtech is developing a product named PPP, which is processed into a powder form through enzyme treatment, filtration, and freeze-drying. Furthermore, although PPP remains a proprietary material pending commercialization, we acknowledge the reviewer's concern regarding replicability. Therefore, considering this, we have decided to disclose the peptide sequence analysis data of PPP exclusively to reviewers.

Comment 1C. There may be a difference between the PPP and its composition after digestion (compare line 268, ref 26). This may suggest that although the preparation investigated in vivo and in vitro is the same, the substances causing the observed effects are not the same. I would suggest more caution with cause-effect statements in this paper.

Response: We appreciate the reviewer's kind indication. In the revised manuscript, we carefully deleted that sentence to avoid any misunderstandings.

Comment 1D:

“In this study, porcine placenta peptide increased the concentration-dependent levels of Gpx-1 mRNA, which were reduced by UVB exposure”

  1. Such sentence suggests this is a single peptide (if read separately), which is not studied in this work (a mixture is used).

Response: We appreciate the reviewer's kind indication. As per the reviewer's comment, all content in the manuscript was changed from “porcine placental peptide” to “porcine placental peptides.”

  1. Please either provide reference for the sentence in lines 64-66, or modify it to relate it to the results of this work.

Response: We appreciate the reviewer's kind suggestion. As per the reviewer's comment, we added the reference to the sentence in line 66.

  1. Please check the dose dependence in line 193.

Response: We appreciate the reviewer's kind indication. Our results showed a low tendency to decrease concentration-dependently, so we replaced the sentence according to the reviewer’s comment as following:

Lines 192–194: “PPP inhibited collagenase activity and MMP-1 expression in HDF cells irradiated with UVB (Figure 5B and C).”

  1. In figure 6B-D, the effect of 25 μg/mL dose and higher doses needs to be addressed, as it is not straightforward.

Response: We appreciate the reviewer's kind comment. Based on reviewer comments, we noted that 50 and 100 μg/mL of PPP were the concentrations at which we observed enhanced mRNA expression of factors related to skin hydration.

Lines 212–214: “Additionally, the mRNA expression of the factors associated with skin hydration was significantly increased at PPP treatment concentrations of 50 and 100 μg/mL (Figure 6B–D).”

  1. Please consider modification in line 349 – inflammation (negative effect) is next to positive effects, which may suggest that the preparation in increasing inflammation (the direction of regulation).

Response: We appreciate the reviewer's kind suggestion. We divided the sentences into negative and positive effects and modified them as follows:

Lines 346–349: “These peptides inhibited wrinkle formation and inflammation by attenuating the phosphorylation of p38 MAP kinase and JNK and increasing the levels of factors related to moisturization and antioxidants.”

  1. Line 351: are the preclinical studies referring to this work, or other results? This should be specified and, if the preclinical character is assigned, mentioned in the text earlier.

Response: We appreciate the reviewer's comment. As consideration of the reviewer's comment, We carefully revised the sentences in the Discussion section as follow:

Lines 349–351: “These results suggest that porcine placental peptides have excellent wrinkle improvement and moisturizing effects and have potential as a functional food ingredient.”